# The Effect of a Sleep Intervention on Sleep Quality in Nursing Students: Study Protocol for a Randomized Controlled Trial

**DOI:** 10.3390/ijerph192113886

**Published:** 2022-10-25

**Authors:** Cayetana Ruiz-Zaldibar, Beatriz Gal-Iglesias, Clara Azpeleta-Noriega, Montserrat Ruiz-López, David Pérez-Manchón

**Affiliations:** 1Department of Nursing, Faculty of Health and Education, University of Camilo José Cela, 28692 Madrid, Spain; 2Department of Medicine, Faculty of Health and Biomedical Sciences, Universidad Europea, 28670 Madrid, Spain

**Keywords:** nursing students, sleep quality, sleep hygiene, nursing, health education, programme, intervention

## Abstract

We develop a protocol for assessing the impact of an intervention aimed at improving sleep quality among university nursing students. The study is designed as a pilot randomized controlled trial to be applied during the 2022-23 academic year and is registered at Clinical Trials Gov website (NCT05273086). A total of 60 nursing students will be recruited from a Spanish university. They will be divided into two groups: (30) intervention group and (30) control group. The intervention group will attend two cognitive–behavioural therapy sleep programme sessions focused on knowledge of anatomical structures involved in sleep, chronotype, synchronization, and good sleeping habits. Subjective and objective sleep quality will be assessed before and after the intervention for both groups. In addition to sleep quality, socio-demographic parameters, physical activity, lifestyle habits, and anthropometric measures will be considered prior to intervention. Finally, a satisfaction questionnaire will be applied for posterior analysis. This study is an innovative, relevant intervention that aims to improve sleep quality among university nursing students. Both the approach and the use of objective and subjective validated outcome measurements are key features of this study.

## 1. Introduction

Sleep has been defined as an essential brain state for maintaining energy and restoring bodily function [1]. Sleep quality has a huge impact on health [2], and it is considered a leading public health problem [3]. Although there has been interest in sleep quality in different disciplines such as nursing, medicine, and psychology, it has not yet been clearly defined from a neurophysiological point of view. The concept analysis by Nelson et al. [4] shows that sleep quality is determined by five components: (1) sleep efficiency, total sleep time-total time in bed ratio; (2) sleep disturbance; (3) sleep latency, defined as the time taken to pass from wakefulness to sleep; (4) sleep duration in 24 h; (5) and waking after sleep onset, or in other words, the total time awake from sleep onset to finally waking up.

Good sleep quality is a predictor of physical and mental health, well-being, and overall vitality [5]. However, poor sleep quality, determined by a negative subjective perception of sleep, sleep onset times, short sleep duration, and difficulties combining sleep and daytime activity [6], has been related to psychiatric disorders such as depression, anxiety, and cognitive difficulties, with reduced physical health, premature ageing, and lower efficiency at work [7].

University students are one of the population groups that suffer most sleep-related problems. Both lack of sleep and sleep disorders are the main characteristics involved in low sleep quality among university students [8]. Over time, it has been noted that the weekly sleep pattern of this group is determined by few hours sleeping during the week and recovering sleeping hours at the weekend [7,9]. This deregulation causes them to experience delayed sleep phase syndrome [10], which is characterised by difficulty falling asleep at a socially acceptable time of night and inability to wake easily in the morning [11]. This syndrome has been related to poor academic performance, behavioural, emotional, and psychological problems, as well as increased drug and alcohol consumption [12]. 

Nursing students face complex situations that can specifically affect their sleep quality [13]. Different factors have been identified as limiting the quality of sleep of this group, especially during the first year of professional training [14]. There were individual psychosocial factors [15,16,17], regarding the academic area and professional environment where they describe the possibility of making mistakes in care practice and handling equipment, as well as lack of knowledge and professional skills [18]. 

Several studies have implemented intervention programmes to improve sleep quality among the adult population without resorting to the use of drugs [19]. However, the published literature among the university student population with no prior sleep-related pathologies is scarce, despite the fact that they are considered a high-risk group. To the best of our knowledge, only one systematic review [20] and a meta-analysis [21] that address interventions to improve sleep in university students have been published so far. The latter includes 14 studies, of which six were conducted with university students and the rest with young students and adolescents. 

Many interventions in both the general population and university and adolescent populations focus on treating insomnia and mental health problems such as anxiety and depression. These works have a practical approach to sleep quality, but they do not take into account fundamental elements such as circadian rhythms, motor activity, skin temperature, sleep phases and position in 24 hours, which are key to understanding sleep quality and being able to improve it [22,23,24].

Interventions to improve sleep quality generally have several fundamental cornerstones. First, students are unaware of how sleep deprivation or disturbance affects cognitive functioning [10]. This justifies educational interventions to understand how the brain regulates sleep function. Despite some studies suggesting that people suffering insomnia who receive theoretical sleep training can significantly improve sleep quality [25], the relationship between knowledge and better sleep quality is unclear [10]. Thus, interventions should have a multi-factor approach including different elements that provide attendees with tools to improve their sleep. 

Several studies indicate that cognitive-behavioural therapy yields positive results in this type of intervention [26] as it involves improving sleep by modifying mindset and behaviour patterns. Therapy has different types of treatment that can be used individually or in combination, such as relaxation, stimulus control, restricting sleep and sleep hygiene [27]. Cognitive–behavioural therapy has a multi-component nature, so it can be used to improve health in relation to sleep quality when there is no associated pathology [20]. This is the case of specific sleep hygiene and stimulus control techniques related to sleep [26] which focus on health. Sleep hygiene appears to be a key factor in these interventions as it addresses a series of behavioural and environmental recommendations aimed at promoting healthy sleep from a psychoeducational perspective. During education on sleep hygiene, users learn about healthy sleep habits, which is a first-line intervention for promoting sleep health [26,28].

Assessing interventions to improve sleep quality is another issue that needs to be addressed. The complex nature of the construct requires the use of objective (external and metric assessment) and subjective (perception of sleep quality) aspects in its assessment. Polysomnography and actigraphy are two different methods used to measure sleep quality [4]. For a subjective sleep assessment, the *Pittsburgh Sleep Quality Index, PSQI* [20] is typically used. Polysomnography is the gold standard method [29], but its use is limited due to difficult availability and home monitoring, combined with the lack of other validated outpatient tools that can replace this test [30]. Studies often use actigraphy to assess sleep [21] because it has proven to be sensitive in its measurement, but again it is of limited use for certain parameters, such as wakefulness during the night. This means that it is not a complete measure for assessing sleep quality and cannot be considered validated like polysomnography [24]. There are no studies using both measures as complementary measures to obtain data on sleep quality in adults [19], and more specifically in young people and adolescents [26]. In fact, most of the systematic review studies mentioned above use PSQI or actigraphy, translating the results into total hours of sleep [20,21]. This leads to a partial and biased assessment of how these interventions affect sleep quality.

Lastly, despite numerous studies analysing the factors associated with poor sleep quality in nursing students [7,31], no intervention study was found in the literature to design, implement, and assess an interventional programme to improve sleep quality in this group.

The aim of this study is to develop a protocol for assessing the impact of an intervention aimed at improving sleep quality among nursing university students.

## 2. Materials and Methods

The study protocol is based on a randomized controlled trial. Participants will be randomly assigned to a control and an intervention group. Only participants in the latter group will receive the intervention based on two cognitive–behavioural therapy sleep programme sessions.

### 2.1. Hypotheses

We pose the conceptual hypothesis that participants in the intervention group will improve their sleep quality, measured by objective monitoring parameters, and their subjective perception. 

### 2.2. Design

The study will be conducted at a Spanish university as a two-arm pilot randomized clinical trial (intervention and control). It will be single-centre and single-blind (data analysts), comparing two conditions: the intervention group (IG), with a specific intervention on sleep quality, and the control group (CG), with no intervention. Figure 1 shows the study flow chart. The random clinical trial report will follow the recommendations for interventional trials (SPIRIT) [32]. This protocol is registered at ClinicalTrials.gov (accessed on 15 September 2022) with reference number NCT05273086. 

### 2.3. Sample/Participants

The study will be conducted at a university in Madrid (Spain). The intervention environment will be exclusively affecting undergraduate students during the first term of the 2022-23 academic year (October 2022–February 2023).

#### 2.3.1. Inclusion Criteria

Students aged 18 to 25 years.Students enrolled in their first full year of a nursing degree, 2022-23.

#### 2.3.2. Exclusion Criteria

Prior mental pathology and/or sleep disorder diagnosis with or without medication (hypnotics, sedatives, and melatonin).Participants that use any drugs to sleep.Combining work and studying.

#### 2.3.3. Sample Size Calculation

For a population size of 70, at least 60 samples are required to have a confidence level of 95% so that the real value is within ±5% of the measured/surveyed value. For two-group comparisons, at least 60 samples are required for an expected medium effect [33].

#### 2.3.4. Randomization

An independent researcher who is unaware of the study characteristics will conduct the random assignment procedure. A computer-generated random number sequence (Sealed Envelope™ software version 1.21.0, London, UK), assigned to participants’ academic record number will be used to randomly designate participants to one of the two groups (IG or CG). Random permuted blocks will also be used to reduce the random sequence predictability and guarantee a 1:1 ratio. This sequence will be password-protected in a table and will be hidden from other researchers during the study.

#### 2.3.5. Blinding Design

The study design does not allow the treatment assignment to be blind for the sleep experts facilitating the intervention and the participants. However, analysts will be blinded to the group-assigned treatment. To prevent interobserver variability bias, measurements will be taken by the same researcher in all cases.

### 2.4. Intervention

The intervention group will receive the programme sessions based on knowledge and cognitive–behavioural therapy to improve sleep quality, and the control group participants will continue with their normal routine. This programme will be developed with an active-constructive learning methodology in which students interact with the material so that their commitment to learning generates deeper knowledge [34]. For this reason, the intervention group will be subdivided into two groups of 10 participants. Reducing the group numbers in this way will enable greater participation and skills acquisition to improve sleep quality. This includes acquiring notions of chronobiology; locating nervous structures involved in biological rhythms, awareness of the existence of biological and social stimulus that determine human circadian rhythms and the importance of their synchronization, self-awareness of the chronotype; knowledge of variables that determine good sleep quality, knowledge of one’s own sleep pattern, routines that improve sleep quality, and resources to avoid variables that are detrimental to sleep quality.

The intervention will consist of face-to-face 90-min group sessions executed twice a week during November and December 2022. Each session will be divided into four parts focused on different aspects related to sleep quality. The first session will focus on knowledge of anatomical structures involved in sleep and their effect at a cognitive level. Students may also self-assess their sleep pattern so that strategies can be adapted to improve this pattern at the next session. The second session will focus on developing skills for better sleep hygiene, working on the students’ individual sleep routines and the results of their questionnaires and self-reports so that the elements of sleep hygiene can be adapted to each case. Table 1 specifies the objectives, session activities, required materials, and duration of the intervention.

The experts implementing the intervention will be a PhD candidate in biology specialized in the brain function of sleep, chronobiology, and circadian rhythms, and a registered nurse expert in health promotion and in healthy lifestyles education, who will conduct the particular part of the sessions related to sleep hygiene and nocturnal habits. 

### 2.5. Data Collection

The assessment process will include self-assessment questionnaires in which participants must answer questions on their perception related to sleep quality, socio-demographic aspects and lifestyle. 

#### 2.5.1. Dependent Variables

Sleep quality: Kronowise 3.0, Kronohealth, S.L., Spain. Kronowise 3.0 is a wrist device that conducts ambulatory circadian monitoring (ACM) based on thermometry, motor activity, and body position (TAP). TAP uses a combination of sensors in an algorithm that has been validated as ambulatory polysomnography. It was validated by comparing the assessment of these parameters with polysomnography in adults [35] and in patients diagnosed with Parkinson’s [36]. The device can measure sleep quality and circadian rhythms and identify circadian chronodisruption through parameters such as sleep latency, total sleep time (in minutes), sleep efficiency, number of awakenings and time between waking and sleep [35]. It can also identify sleep phases, nocturnal awakenings, temperature, exposure to infrared and blue light as well as other parameters such as time in bed, sleep onset, awake time, sleep interval, total time in movement, time in movement index, sleep acceleration index, wrist sleep temperature, napping time and napping frequency [36]. Participants will wear the device for seven consecutive days, including weekdays and weekends. Both groups will be measured twice (before and after the intervention), meaning that all participants will wear the device for a total of 14 days. Participants will be given appointments to fit and remove the devices. Information provided by the devices will be entered into the software that will be sent to Kronohealth S.L., Spain, which will produce reports of both measurements with the results of the parameters mentioned above, with the sleep quality indices established according to the device validation [35].Perceived sleep quality: Pittsburgh Sleep Quality Index, PSQI [37]. The PSQI contains 19 items and seven clinically important components related to sleep quality: subjective sleep quality, sleep latency, sleep duration, sleep efficiency, sleep disturbances, use of sleeping medication, and daytime dysfunction. Responses will be reported based on a Likert-type scale from 0 to 4. Overall sleep scores of 5 or less will be considered good quality, whereas scores of 5 or more will be considered low quality. The PSQI version that will be used is validated into the Spanish university framework by De la Vega et al. [37] with a Cronbach’s alpha of 0.72. The questionnaire will be given to all participants twice (before and immediately following the intervention).

#### 2.5.2. Independent Variables

All independent variables will be assessed before the intervention, except for the intervention group’s satisfaction questionnaire, which will be measured immediately afterward. Questionnaires on socio-demographic data and toxic habits are prepared ad hoc by the researchers.

*Socio-demographic data*: Date of birth, age in years, gender and sleep habits during the study (alone or accompanied).*Toxic habits*: Tobacco consumption (cigarettes/day), alcohol consumption (daily, weekly, sporadic, glasses), drug consumption (yes/no and type), and stimulating drink consumption (coffee, sugary drinks, energy drinks).*Physical exercise*: The International Physical Activity Questionnaire (IPAQ), simplified version [38]. The questionnaire consists of seven questions on frequency, duration and intensity of activity (moderate and intense) in the last seven days, as well as walking and sitting time during a working day. The questionnaire is classified into the following levels: low, moderate, and high physical activity. The higher the score, the more physically active the profile. The questionnaire is validated with a mean reliability of 0.80 [38].*Anthropometric variables*: Participants will report their weight (kg) and height (cm).*Satisfaction with the programme:* Using a questionnaire adapted to the proposal by Azpeleta et al. [39] with 10 questions and responses based on a Likert-type scale from 1 to 10. Greater satisfaction will be awarded a higher score.

Table 2 shows the data collection and monitoring instruments.

### 2.6. Ethical Considerations

This study will be conducted according to the principles established in the Declaration of Helsinki, in the Convention on Human Rights and Biomedicine (Oviedo Convention), and in the UNESCO Universal Declaration on the Human Genome and Human Rights. This study has been approved by the Research Ethics Committee of Camilo José Cela University (code: 06-22-UCJC-Sleep). 

All participants will be informed of the duration and characteristics of the study, and of its voluntary nature. After receiving a detailed explanation of the project, any doubts raised by participants will be answered before the students are asked to sign the informed consent form, which mandatorily must be signed to participate.

Participants may abandon the study at any time, with no further consequences. The highest professional conduct and absolute confidentiality will be always maintained, in accordance with European Regulation (EU) 2016/679 on the protection of natural persons and data processing and free movement, the Framework Act 3/2018 on personal data protection and the guarantee of digital rights, and Act 14/2007 on biomedical research. Only the principal researcher of the project will have access to the codified database.

### 2.7. Study Procedure

Participants will be recruited over two weeks and via four channels. Information and informed consent that is mandatory for participation will be provided on the institution’s mobile app, in an institutional email and through two classroom sessions. Furthermore, information will be provided on informative posters on the electronic boards placed around the university premises.

After the two-week recruitment phase, all informed consent forms will be collected, and the inclusion/exclusion criteria applied. This will be conducted by a team researcher who will not take part in the intervention. Once the recruitment phase has been completed, students will be codified with their personal academic record number, and the intervention and control group will be randomly assigned. The intervention group will be divided into two groups (intervention group 1 and intervention group 2) according to session schedule preferences.

As shown in Figure 2, the following phases consist of the pre-intervention assessment for both groups, two behavioural–cognitive sessions with the intervention groups, and the post-intervention assessment. 

Ten Kronowise 3.0 devices will be acquiring data a whole week for each student, so the different phases will be staggered for the groups. This means that the intervention protocol will last a total of 16 weeks (see Figure 2).

### 2.8. Data Analysis

For the statistical analysis, all variables will be coded by using IBM Statistics SPSS v.21 software (Chicago, USA) for Windows and revised twice. A protocol approach will be followed where missing data will not be included in the analysis.

All descriptive statistical parameters will be expressed as absolute (‘n’) and relative (‘%’) frequencies for each qualitative variable category. Quantitative variables will be analysed with the mean, median, and standard deviation to observe their behaviour at a confidence interval of 95%. Normal variable distribution will be analysed by using the Kolmogorov–Smirnov test with Shapiro–Wilk correction. 

A repeated measures ANOVA will be used to compare mean differences between the intervention and control groups over time, considering the time–group interaction effect. The size of the effect will be analysed based on Cohen’s d [40]: 0–0.3 low, >0.3–0.8 moderate, and >0.8 large effect size. Any incomplete questionnaires will be excluded from the analysis. Results will be considered statistically significant when *p* < 0.05.

The PROCESS macro will be used to examine the mediating effects of sleep quality on the relationship between attending the sleep intervention or not. This analysis uses linear regression to estimate indirect effects according to Hayes and Rockwood’s methods recommended for clinical studies and focuses on two measurement moments [41]. Post-treatment scores will be added as mediators, and baseline scores for the outcome variables and mediators will be added to the models as covariates. Least-square path analysis will be used and the bootstrap confidence interval (5000 permutations) will be applied to estimate indirect effects.

## 3. Discussion

This research aims to promote healthier lifestyles in nursing undergraduate students. It is specifically addressed to target groups that share risk factors that can compromise sleep quality and to act at the educational level. 

The main strength of this paper is the intervention itself, which is based on cognitive–behavioural therapy and improved understanding of the physiology of sleep. The physiology of sleep targets both the nervous structures involved in sleep and the self-knowledge of sleep patterns and sleep hygiene strategies, which are the cornerstones of the intervention [42]. It combines aspects to promote a cognitive process that consolidates the development of everyday life skills, providing positive patterns for healthy sleep. This intervention may be the first step to develop future studies with application in nurses to improve sleep disorders or work–life balance [43].

The assessment consists of objective and subjective components for a whole comprehensive approach to improve sleep quality. More specifically, the use of a TAP tool, validated as an ambulatory test using a wrist device vs. polysomnography, is an innovative added value of this study, as it will provide reliable, complete results on how the intervention affects sleep quality [35]. 

Lastly, the methodological design of the controlled clinical trial is crucial in this type of study to guarantee results on the effectiveness of the intervention. 

Aside from the strengths already mentioned, other limitations should also be noted. First, the sample size may fall short of capturing differences between groups. However, the lack of prior studies and the complexity of developing the study depending on available devices, means that obtaining pilot study results will be enriching to analyse intervention barriers and facilitators for large-scale future implementation. 

Additionally, we cannot guarantee a possible type two error, as randomization will be individual, and there may be contamination between the control and intervention groups as they interact during their studies.

Last, Kronowise 3.0 devices are highly useful for assessing the study despite their limited availability and cost. However, developing this pilot study will provide preliminary data which is crucial to apply for national or international external project funding to sustain the cost involved for interventions on a larger scale. 

## 4. Conclusions

The methodological design of this study supports interventions that can favour better sleep quality in university nursing students. The intervention is a novel strategy combining cognitive learning about sleep as a physiological process, the use of technological devices to monitor sleep parameters, and addressing students’ knowledge about their individual habits through cognitive behavioural therapy. The research to be conducted using this protocol aims to demonstrate the positive effect of such an intervention on sleep quality in nursing students. 

## Figures and Tables

**Figure 1 ijerph-19-13886-f001:**
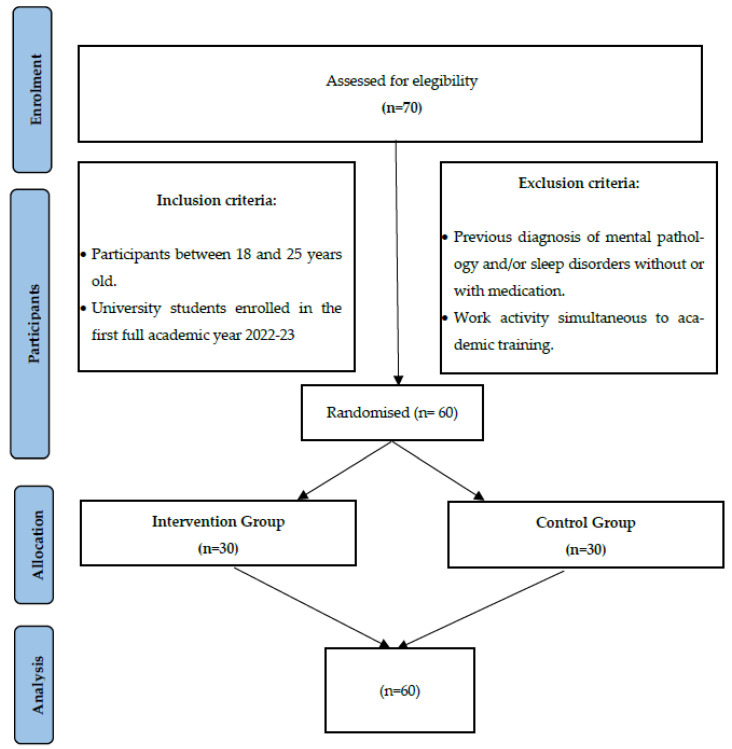
Participant’s flowchart diagram.

**Figure 2 ijerph-19-13886-f002:**
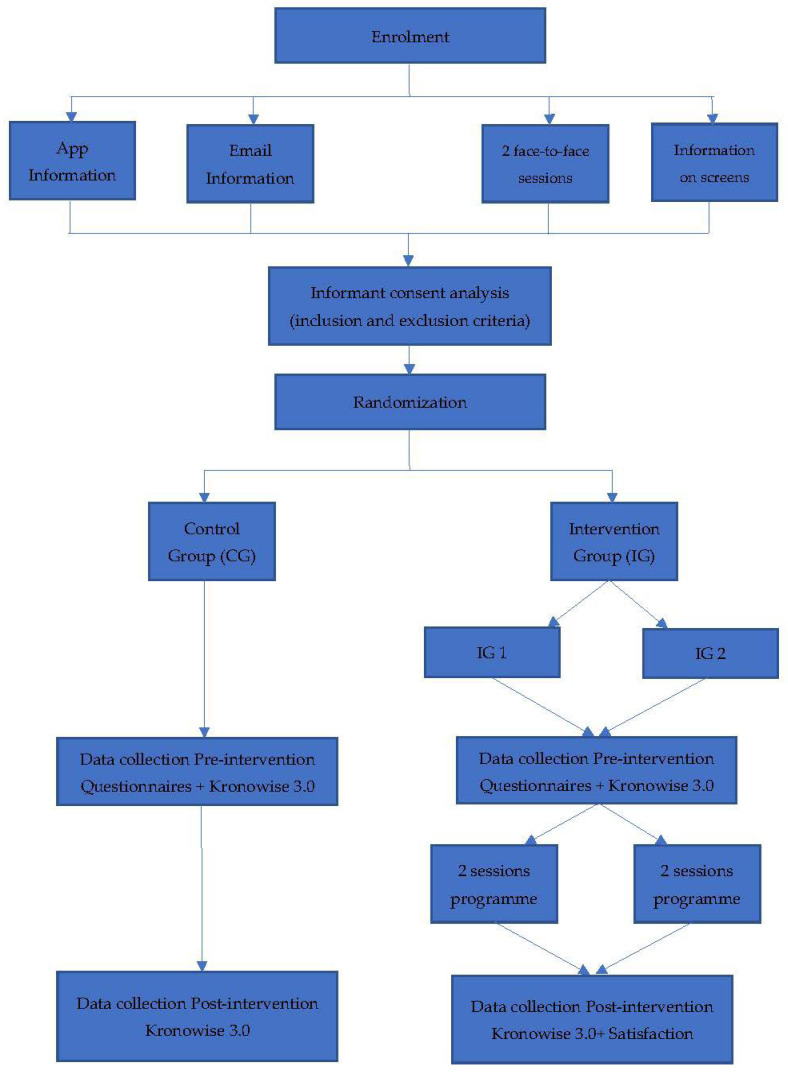
Study procedure.

**Table 1 ijerph-19-13886-t001:** Distribution of the sessions of the sleep quality programme for nursing students.

Aims	Content	Material	Time
First session			
Acquire knowledge of how sleep works	Explanation of the concepts of chronobiology, biological rhythms in humans, internal clock and synchroniser, and the importance of structures such as the hypothalamic suprachiasmatic nucleus (SCN) or the pineal gland, and the melatonin synthesised by it in these biological rhythms.	Digital anatomical table explaining the different anatomical structures of the body involved in sleep.	30 min
Acquire knowledge of the anatomical structures involved in sleep	Localisation of the structures mentioned in diagrams and anatomical models of the nervous system.	Analysis of the anatomical structure of the brain.	10 min
Acquire knowledge about sleep patterns	Addressing the concept of chronotype and the existence of three times that affect human circadian rhythms (biological, artificial and social), and how a mismatch between them leads to a state called chronodisruption.	Digital screen presentation with summary of main. concepts	20 min
Self-awareness of sleep pattern; analysis of one’s own sleep behaviour	Completion of two self-knowledge questionnaires on sleep patterns: Morning/early morning test and three-times test. Group discussion on results.	Web Cronolab: https://www.um.es/cronobiologia/taller-del-relojero/autoevaluacion/ (accessed on 19 September 2022)	30 min
Second session			
Self-awareness of sleep pattern; analysis of one’s own sleep behaviour	Completion of a questionnaire on daytime sleepiness. Group discussion of results.	Web Cronolab: https://www.um.es/cronobiologia/taller-del-relojero/autoevaluacion/ (accessed on 19 September 2022)	10 min
Group discussion on sleep related behaviours and lifestyle habits; acquisition of healthy lifestyle habits that improve sleep	Sleep hygiene: avoid caffeine, avoid nicotine, avoid alcohol, exercise regularly, manage stress, reduce bedroom noise, sleep timing regularity and balanced diet.	Recommendations of the Spanish Sleep Society for good sleep and the circadian system. Spanish Sleep Society website: https://ses.org.es/ (accessed on 19 September 2022)	30 min
Group discussion on sleep habits and behaviours; acquiring healthy sleep habits	Sleep behaviour: establishing regular bedtimes and wake-up times, avoiding lying in bed waiting to fall asleep, going to bed only when sleepy, maintaining predictable activities before bedtime, establishing a standard wake-up time, creating a routine for waking up quickly, leaving the bed or bedroom after long periods of wakefulness, and avoiding behaviours incompatible with sleep in bed or the bedroom.	Recommendations of the Spanish Sleep Society for good sleep and the circadian system. Spanish Sleep Society website: https://ses.org.es/ (accessed on 19 September 2022)	20 min
Self-awareness of modifiable sleep hygiene behaviours	Delivery of self-reports with application of sleep hygiene and specific sleep behaviour. Group discussion with comparison of reports of real patients with pathologies.	Review of reports of patients with sleep disorders and review of self-reports	30 min

**Table 2 ijerph-19-13886-t002:** Data collection and follow-up instruments.

	Pre-Intervention		Post-Intervention	
	CG	IG	CG	IG
Socio-demographic data	●	●		
Toxic habits	●	●		
IPAQ	●	●		
Anthropometric variables	●	●		
Satisfaction with the programme				●
Kronowise 3.0	●	●	●	●
PSQI	●	●	●	●

CG: control group; IG: intervention group.

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
