# Peer review of "The Effect of a Sleep Intervention on Sleep Quality in Nursing Students: Study Protocol for a Randomized Controlled Trial"

_ijerph, 2022, doi:10.3390/ijerph192113886_

Round 1
Reviewer 1 Report
Thank you for the opportunity to review the manuscript entitled “The effect of a sleep intervention on sleep quality in nursing students: study protocol for a randomized controlled trial”. The manuscript is well-written, and the proposed methodology is rigorous.
1. The literature review here appears relatively thin and it’s unclear what is so particularly unique about this new method. I would like the authors to enhance the theory behind why this new intervention will be useful and propose more sophisticated analyses (e.g., mediation, moderation), so that we can learn more about how the proposed intervention will work.
2. There are some small number issues (exclusion has the same number heading as inclusion) and the headers do not need colons.
Author Response
Thank you very much for your comments.

Reviewer 2 Report
The present paper consists in the description of a study protocol proposal aimed at evaluating the effects of a series of intervention including cognitive-behavioural therapy sleep programme sessions focused on knowledge anatomical structures involved in sleep, chronotype, synchronization and good sleeping habits in nursing students.
My main comments are as follow:
Introduction
The introduction is too long and of some extent repetitive.
It should be globally shortened by eliminating the parts that are not essential for the rational of the proposed protocol.
Lines 131-134: this part should be moved into methods session
Material and methods
The sample size calculation should be reported anyway, by using previous similar studies, as well as the number of female and male at the enrolment.
To increase the power of the study why not starting from people identified at the enrolment as a “poor sleepers” based on criteria used by previous studies? In the present protocol it seems that also subjects that do not suffer for sleep disturbances will be treated.
I suggest avoiding patients taking any drug for reducing the confounding effects.
Line 176: “Having signed the informed consent form” is not a criterion of inclusion, however it is mandatory to participate in the study and must be obtained and reported.
Line 181 Sample size calculation, see above.
A general daily routine should be followed by the enrolled subjects during the different phases of the study. For example, too much or too low physical exercise performed during the evaluation of sleep characteristics may influence the results.
Line 238 this way to assess sleep should be specified. What is “the objective assessment of sleep”? What is the “ambulatory device”?
Line 245: add citation, there are other references on the application of such a device in previous (or similar) studies?
Lines 247-249: this sentence is generic: how the device can “identify different pathologies”, please explain or described what parameters will be provided by the device proposed.
2.7 Study procedure: I have concern on the methodology to recruit the participants, this may represent a bias due to
Line 342: two: in the same sentence should be avoid, please rewrite the sentence.
Table 2: why the Kronowise 3.0 has not be included in the post intervention phase? It is probably a refuse?
The Kronowise 3.0 is reported in the flow chart of Figure 2.
Author Response
Thank you very much for your comments. We have changed the manuscript and response to your suggestions in the attached document.

Round 2
Reviewer 2 Report
Thank you for providien changes in the manuscript as suggested. Please address the additional minor comments.
1. I suggest to keep "The aim of this study is to develop a protocol for assessing the impact of an intervention aimed at improving sleep quality among nursing university students" (lines 116 an 117) at the end of Introduction ad starts Materials and Methods by a descriprion of the case control randomized study.
2. Consider adding the following references:
Int J Environ Res Public Health. 2021 Nov 23;18(23):12283. doi: 10.3390/ijerph182312283
Int J Environ Res Public Health. 2018 Sep 18;15(9):2038. doi: 10.3390/ijerph15092038.
Author Response
Thank you for the opportunity to improve our document with your comments.
|
Comment |
Answer |
|
1. I suggest to keep "The aim of this study is to develop a protocol for assessing the impact of an intervention aimed at improving sleep quality among nursing university students" (lines 116 an 117) at the end of Introduction ad starts Materials and Methods by a description of the case control randomized study |
Thank you for your appreciation. We have moved this part to the end of the introduction. The material and methods part begins with a brief explanation of the randomized controlled trial study. |
|
2. Consider adding the following references: Magnavita N, et al Int J Environ Res Public Health. 2021 Nov 23;18(23):12283. doi: 10.3390/ijerph182312283 Shiffer D, et alInt J Environ Res Public Health. 2018 Sep 18;15(9):2038. doi: 10.3390/ijerph15092038.
|
Thank you very much for the references. These articles are very interesting for our research. We have added the research related to nurses because we consider that it fits better in the discussion. Related to this, we are very interested in following the next steps with nursing professionals, so this reference will be very useful for our future research. |